# Early Inoculation of Microbial Suspension in Suckling Piglets Affects the Transmission of Maternal Microbiota and the Associated Antibiotic Resistance Genes

**DOI:** 10.3390/microorganisms8101576

**Published:** 2020-10-13

**Authors:** Caroline S. Achard, Veronique Dupouy, Laurent Cauquil, Nathalie Arpaillange, Alain Bousquet-Melou, Nathalie Le Floc’h, Olivier Zemb

**Affiliations:** 1Génétique Physiologie et Systèmes d’Elevage (GenPhySE), Université de Toulouse, Institut National De Recherche Pour L’agriculture, L’alimentation Et L’environnement (INRAE), Institut National Polytechnique de Toulouse (INPT), École nationale vétérinaire de Toulouse (ENVT), F-31320 Castanet Tolosan, France; cachard@lallemand.com (C.S.A.); laurent.cauquil@inrae.fr (L.C.); 2Lallemand SAS, 19 rue des Briquetiers, BP 59, 31702 Blagnac CEDEX, France; 3Innovations thérapeutiques et résistances (INTHERES), Université de Toulouse, INRAE, ENVT, F-31300 Toulouse, France; v.dupouy@envt.fr (V.D.); n.arpaillange@envt.fr (N.A.); a.bousquet-melou@envt.fr (A.B.-M.); 4Physiologie, Environnement et Génétique pour l’Animal et les Systèmes d’Élevage (PEGASE), Institut national de recherche pour l’agriculture, l’alimentation et l’environnement (INRAE), Institut Agro, 35590 Saint-Gilles, France; nathalie.lefloch@inrae.fr

**Keywords:** competitive exclusion, microbiota, piglet, antibiotic resistance genes

## Abstract

Antibiotic resistance of microbes thriving in the animal gut is a growing concern for public health as it may serve as a hidden reservoir for antibiotic resistance genes (ARGs). We compared 16 control piglets to 24 piglets fed for 3 weeks with S1 or S2 fecal suspensions from two sows that were not exposed to antibiotics for at least 6 months: the first suspension decreased the erythromycin resistance gene *ermB* and the aminoglycoside phosphotransferase gene conferring resistance to kanamycine *(aphA3)*, while the second decreased the tetracycline resistance gene *tetL*, with an unexpected increase in ARGs. Using 16S RNA sequencing, we identified microbial species that are likely to carry ARGs, such as the lincosamide nucleotidyltransferase *lnuB*, the cephalosporinase *cepA*, and the tetracycline resistance genes *tetG* and *tetM*, as well as microbes that never co-exist with the tetracycline resistance gene *tetQ*, the erythromycin resistance gene *ermG* and *aphA3*. Since 73% of the microbes detected in the sows were not detected in the piglets at weaning, a neutral model was applied to estimate whether a microbial species is more important than chance would predict. This model confirmed that force-feeding modifies the dynamics of gut colonization. In conclusion, early inoculation of gut microbes is an interesting possibility to stimulate gut microbiota towards a desirable state in pig production, but more work is needed to be able to predict which communities should be used.

## 1. Introduction

The prevalence of antibiotic resistance genes (ARGs) in livestock is country-specific. For example, Chinese pigs have more ARGs encoding resistance to chloramphenicol, gentamycin B, kanamycin and neomycin than their French and Danish conspecifics [1]. Country-specific ARG profiles also exist within Europe, where microbiota of pigs from Italy have more of the streptomycin-resistant mutation *strA* than those from France, Denmark or Sweden [2]. Furthermore, the country-specific differences of ARGs in livestock seem to penetrate the human gut. Indeed the ARGs in Spanish, Danish and U.S. citizens mirror the antibiotics used in livestock in each of these countries, such as streptomycin-resistance which is more prevalent in Spain than in Denmark [3]. More indirect evidence comes from the nisin-producing operons in bacterial isolates from human, porcine and bovine hosts, revealing extensive bacterial transmission between humans and pigs [4]. ARGs in livestock may therefore become a public health issue if they are transferred to pathogenic bacteria that inhabit the human gut.

In response to the increase in antibiotic resistance, the European Union banned antibiotics as growth promoters for livestock in 2006, and data suggest that the impact of this ban on the performances of growing pigs is relatively minor at the country-level in Denmark [5], or in carefully controlled groups [6,7]. However, a long-term study conducted at the University of Kentucky showed that stopping antibiotics could also be detrimental to reproductive performance since sows not treated with antibiotics farrowed and weaned fewer pigs per litter [8]. Therefore, excluding antibiotics from the nursery is far from trivial.

As a matter of fact, the 2006 ban on fattening animals might not be sufficient to decrease antibiotic resistance, as seen by Spanish Methicillin-resistant Staphylococcus aureus strains that became more resistant to multiple antibiotics between 2009 and 2018 [9]. Indeed, ARGs may incur almost no fitness cost for the bacteria. For example, *Campylobacter* spontaneously evolved a resistance against macrolides and was not outcompeted by its sensitive kin in an antibiotic-free environment [10]. This lack of fitness cost implies that ARGs might persist in the commensal microbiota years after the antibiotic treatment [11,12]. In this context, long-lasting resistance genes found in the maternal environment are an issue because young animals are likely to inherit antibiotic-resistant microbes from their mothers and the environment. Indeed, the microbiota of rabbit kits that have been adopted just after birth is closer to the microbiota of the lactating doe than to the microbiota of the biological mother [13], and piglets are sensitive to their surrounding microbes as well [14]. Therefore, long-lasting resistance genes are a logical target to fight against antibiotic resistance inherited by piglets from lactating sows.

The idea of using a complex microbial community to remediate against a specific bacterial species was first described in 1973 [15] to outcompete pathogenic *Salmonella* in broilers. Competitive exclusion was also successfully adapted to the exclusion of antibiotic-resistant *E. coli* in broilers [16,17] and in mice after an antibiotic treatment [18]. Nowadays, human fecal transplantation using complex microbial communities can cure *Clostridium difficile* infections [19], and lyophilization protocols have been developed to facilitate its routine use [20]. However competitive exclusion studies using complex microbial communities remain relatively scarce for mammalian livestock. In pigs, *Salmonella* could be outcompeted using complex cultures grown in vitro [21,22], but using competitive exclusion has been reported to sometimes increase the antibiotic resistance of *E. coli* [23] or to have no effect in vivo [24].

In this paper, we investigate the potential of complex microbial communities to stimulate the microbiota of piglets from ARG-carrying sows and, especially, to outcompete bacteria carrying ARGs. The complex microbial communities were orally delivered. We measured antibiotic resistance and transposase gene abundance, *Enterobacteriacae* resistance to tetracycline or sulphonamide, and the taxonomic composition of the fecal microbiota of the piglets before weaning.

## 2. Materials and Methods

The experiment was conducted at the France’s new National Research Institute for Agriculture, Food and Environment (INRAE) Pig Physiology and Phenotyping Experimental Facility (UE3P) in Saint-Gilles (France) in compliance with the ethical standards of the European community (Directive 2010/63 EU). Six Large-White x Landrace sows were selected from the INRAE herd and were inseminated with Piétrain semen. All the sows involved in the experiment and their litter were housed in the same building and were gestating and lactating at the same time. The pigs were housed in farming conditions as authorized by the French regulation on animal experimentation for experimentations conducted with an agronomic purpose. At 106 days of gestation, sows were moved to the farrowing room with natural (windows) and artificial light. They were kept in individual farrowing crates (2 × 2.5 m) equipped with a trough and a drinker. The floor of the farrowing pens was made of slatted plastic. The farrowing crates were equipped with 2 infrared heat bulbs for piglets. The ambient temperature was kept at 24 °C to 25 °C in the lactation rooms with heating and ventilation. Farrowing was induced by an intramuscular injection of prostaglandin F2α (2 mL of Dinolytic, Zoetis, France) on day 114 of gestation. Usual farm practices for newly farrowed piglets were individual identification by tagging, iron injection, tail docking, and castration of male piglets. Cross-fostering, if needed, was performed intra-treatment within 2 days after birth but piglets selected for blood and feces samplings were maintained with their biological mother.

### 2.1. Donor Sows and Preparation of the Microbial Suspensions

Two donor sows (first parity for Sow 206 generating the S2 suspension and third parity for Sow 268 generating the S1 suspension) were selected based on their low abundance of sulfonamide- and tetracycline-resistant *Enterobacteriacae*. On the morning of the days of oral inoculation of the piglets, fresh feces were collected from the sows and 20 g were immediately mixed with 110 mL of anoxic physiological salt solution at 0.9% NaCl in a screw-cap 125-mL bottle equipped with a septum. After vigorous shaking, the suspension was made anoxic by three cycles of vacuum/filling with N_2_. The last cycle ended with an overpressure of 500 mbar. The suspension was then immediately administered to the piglets by force-feeding.

### 2.2. Lactating Sows and Experimental Piglets

Four gestating sows (three were primiparous and one was multiparous) were selected on the basis of their relatively high abundance of sulfonamide- and tetracycline-resistant *Enterobacteriacae* (31% ± 18% and 36% ± 15%). After farrowing, the piglets remained with their biological mother during the experiment. Force-feeding took place every two days from Day 2 to Day 23 after birth by placing 2 mL of the above-described microbial suspension near the soft palate using a plastic syringe. Within each litter, five piglets were force-fed with 2 mL of saline (Control), three piglets received the suspension from donor sow S1, and three piglets received the suspension from donor sow S2. Piglets were weaned at 4 weeks. No antibiotic treatment was administrated either to the sows or the piglets.

### 2.3. Feces Sampling and DNA Preparation

Fresh feces were sampled from the lactating sows and from their piglets by rectal stimulation at Day 14 and at four weeks just before being separated from their mother. The donor sows were sampled 1 and 2 months before the experiment and once at the end. The samples were immediately stored at −80 °C. The DNA was extracted from 50 mg of porcine stool using a bead-beating approach, as previously described [25], and stored at −80 °C.

### 2.4. Analysis of the Antibiotic Resistance Genes via qPCR

The DNA concentration was measured using the Quant-iT TM PicoGreen TM dsDNA Assay Kit, according to the manufacturer’s protocol (Invitrogen) and subsequently diluted to 10 ng/µl. 

High throughput real-time qPCR was performed using the Biomark microfluidic system from Fluidigm (San Francisco, CA, USA) using a 96.96 Dynamic Array™ Integrated Fluidic Circuit (IFC). Pre-amplification of the samples, chip loading and qPCR reactions in nanoliter volumes were performed according to the manufacturer’s protocol. A pre-amplification step was applied to all samples for all primer sets except the 16S rRNA primer set. Briefly, 13 ng of total DNA were submitted to 14 PCR cycles using the PreAmp Master Mix (Fluidigm, San Francisco, CA, USA) and a mix of primers (50 nM final concentration). Pre-amplified samples were diluted five-fold after an exonuclease treatment. The diluted pre-amplified samples and the primer sets were loaded in a 96.96 IFC using an IFC Controller HX (Fluidigm, San Francisco, CA, USA). The Biomark thermal protocol was as follows: a thermal mix step (50 °C, 2 min; 70 °C, 30 min; 25 °C, 10 min); a hot start (50 °C, 2 min; 95 °C, 10 min); 35 cycles of PCR (95 °C, 15 s; 60 °C, 60 s); a final melting phase (60 °C to 95 °C).

The list of primers can be found in Appendix A. The quantity of each gene (in arbitrary units) was extrapolated using a generated standard curve with Fluidigm real-time PCR analysis software (v4.3.1). 

### 2.5. Analysis of the Culturable Enterobacteriacae

Feces were sampled from the donor sows 2 months before the experiment, at Day 14 and at Day 27. Feces from the lactating sows and from their piglets were sampled at Day 14 and Day 27 (weaning). Feces (5 g) from each animal were blended, homogenized in 45 mL of peptone water including 30% glycerol, and stored at −80 °C until use. Ten-fold serial dilutions of the homogenate were prepared, and 100-µl samples of the dilutions were spread on MacConkey agar (Sigma, M8302) in order to obtain ten isolates of *Enterobacteriacae*. The antibiotic susceptibility of *Enterobacteriacae* isolates was tested by plating a 24 h pure culture onto Mueller–Hilton agar alone or supplemented with 16 µg/mL tetracycline or 128 µg/mL sulfamethoxazole, according to the *E. coli* epidemiological cut-off values defined by Eucast [26]. Every test included a resistant and sensitive control strain to certify that the antibiotic was active. The proportion of sulfonamide- or tetracycline-resistant *Enterobacteriacae* was calculated for each animal.

### 2.6. Analysis of the Microbiota Composition Using the 16S rRNA Gene

The V3-V4 region was amplified from diluted genomic DNA with the primers F343 (CTTTCCCTACACGACGCTCTTCCGATCTTACGGRAGGCAGCAG) and R784 (GGAGTTCAGACGTGTGCTCTTCCGATCTTACCAGGGTATCTAATCCT) using 30 amplification cycles with an annealing temperature of 65 °C (an amplicon of 510 bp, although the exact length varies depending on the organisms). This region was chosen because it has proved useful for several studies on the variability of the microbiota in pigs exposed to different surrounding microbes and diets [25,27]. Because the V3 kit of Illumina enables paired 300-bp reads, the ends of each read are overlapped and can be stitched together to generate extremely high-quality, full-length reads of the entire V3 and V4 region in a single run. Each pair-end sequence was assembled using Flash software v1.2.6 [28] with at least a 10-bp overlap between the forward and reverse sequences, allowing 10% mismatch. Single multiplexing was performed using an in-house 6 bp index, which was added to R784 during a second PCR with 12 cycles using forward primer (AATGATACGGCGACCACCGAGATCTACACTCTTTCCCTACACGAC) and reverse primer (CAAGCAGAAGACGGCATACGAGAT-index-GTGACTGGAGTTCAGACGTGT). The resulting PCR products were purified and loaded onto the Illumina MiSeq cartridge according to the manufacturer’s instructions. The quality of the run was checked internally using PhiX, and each pair-end sequence was then assigned to its sample with the help of the previously integrated index using bcl2fastq provided by Illumina. The sequences were submitted to the Short-Read Archive [29] with accession number SRP124929.

The sequences were filtered for quality (length > 150 bp, homopolymers <10, 100% unambiguous bases) and reassigned to each sample using barcodes, which were also trimmed. The resulting sequences were clustered into operational taxonomic units (OTU)s using USEARCH [30] and counted in each sample to create a table of relative abundance of each OTU across the samples, yielding between 12,726 and 18,198 high-quality sequences per sample. The representative sequence of each OTU was classified on the Ribosomal Database Project trainset16, reformatted to be compatible with USEARCHv11 (https://www.drive5.com/usearch/manual/sintax_downloads.html), and the taxonomic affiliations with a confidence level lower than 0.8 were labeled ‘unclassified’. The counts were rarefied at 12,726 sequences using the rarefy_even_depth function from the phyloseq package [31]. 

### 2.7. Statistical Analysis

Statistical analyses were performed using Rstudio software with R (v3.3). The difference between the ratios of resistant *Enterobacteriacae* was modeled by a binomial logistic regression in order to take the small number of strains per sample (<11) into account. The differences in ARGs between donor and lactating sows were evaluated for the genes detected in every sample by a Wilcoxon rank sum test corrected for false discovery rate using the Benjamini-Hochberg procedure. The dissimilarities between the bacterial communities of the sows and their piglets were evaluated by the Bray–Curtis distance after rarefaction by the minimal number of reads per sample (i.e., 12,720). The normalized OTU abundance table was used to discriminate the piglets of the three groups (S1, S2 and control) using principal component analysis and discriminant analysis (DAPC) [32]. Briefly, in DAPC, the samples are described in orthogonal axes using a principal component analysis, and a discriminant analysis is applied on the first components. The significance of the separation between samples is quantified with the built-in a-score function with 1000 simulations. Briefly, the a-score compares the separation achieved to the separations than can be achieved when the groups are randomly attributed. The minimal number of OTUs to discriminate the groups in order to obtain significant separation between S1 and S2 was determined by the stepwise addition of the OTUs that contributed the most to the DAPC until significance, using the a-score function with 1000 permutations, is reached. The neutral model was independently fitted to piglets with the same inoculation and the same mother (i.e., 12 sets) using the script provided by Burns et al. [33], and the resulting goodness-to-fit was analyzed by ANOVA. The impact on the ARGs was evaluated with a Friedman test based on the medians. 

The linear and non-linear relationships between the selected OTUs and ARGs were characterized on a filtered OTU table so that the total count of each OTU across the samples is higher than 1000 (i.e., 0.07% of the total counts), which was merged with the ARG dataset using the sample Id. We then computed the maximal information coefficient index using the application provided with Java SE 1.7.0_02 [34] with the percentage of the records necessary to have data in them for both variables before those variables are compared (cv) and the exponent (α) parameters set to 0.6. To account for the baseline correlation bound to occur between any two vectors, the empirical null distribution of the maximal information coefficient was computed using 1000 random permutations of the relative abundances within the variables. The maximal value of the null distribution was set as the significant threshold for the real dataset, i.e., the dataset before randomization. The relationships were labeled as linear when the regression coefficient was higher than 0.5. They were labeled as non-linear when the regression coefficient was negative, and the nonlinearity coefficient was higher than 0.3. The dataset at 14 days and the dataset at weaning were analyzed separately to avoid relationships due to the impact of age.

## 3. Results

### 3.1. The Donor and Lactating Sows Had Different Resistance Profiles

The prerequisite when attempting a targeted competitive exclusion is that the inoculum is depleted in the target. We checked this prerequisite by measuring the proportion of resistant *Enterobacteriacae* over the total count of *Enterobacteriacae* by plating the feces of the sows on selective medium: the proportion of resistant *Enterobacteriacae* was indeed lower in the donor sows that generated suspensions S1 and S2 (sow 268 and sow 206) than in the four lactating sows (sow 512, sow 709, sow 710 and sow 722) (6 ± 8% vs. 36 ± 15%, Appendix A). We also measured the ARGs by quantitative PCR in six samples from the donor sows and the 12 samples from the lactating sows: unfortunately, ARGs in the donor and the lactating sows were not significantly different when correcting for the false discovery rate (Appendix A), even though some heterogeneity was observed prior the experiment (Appendix A). 

### 3.2. Oral Inoculation of Microbial Suspension Affected the ARGs in Piglet Feces

Two sows that were not exposed to antibiotics for at least 6 months generated the S1 and S2 suspensions that were fed from Day 2 to Day 23 to the piglets born from the four lactating sows: force-feeding the S1 suspension impacted 15 ARGs in the piglets, while force-feeding the S2 suspension impacted eight ARGs in the piglets compared to the control piglets (Table 1 and Table 2, respectively). Some ARGs were impacted at both sampling dates, while others were impacted at one date only. For example, the S1 suspension decreased the abundance of the aminoglycoside nucleotidyltransferase gene *aadE_2* at 14 days, but this effect was no longer observed by the date of weaning (i.e., 4 weeks). It should be noted that *aadE_2* tended to be lower in donor sows than in lactating sows according to a Wilcoxon test (*p* = 0.052, Appendix A). Remarkably, other ARGs such as the glycopeptide resistance gene *vanTG* or *lnuB* were increased by the S1 suspension in both sampling dates (Table 1).

Interestingly, the decreased abundance of a specific ARG in the suspension was not always a guarantee of a decrease in that ARG in the force-fed piglets. The difficulty of predicting the success of the competitive exclusion is striking for the 16S rRNA methyltransferase *npmA* against aminoglycosides, which was present in each lactating sow, undetectable in the S1 and S2 suspensions but not modified by force-feeding the suspensions (*p* = 0.32 according to the Wilcoxon test) (Appendix A). The complexity of predicting the success of the ARG exclusion from the ARGs in the donor and the lactating sow is presented in Appendix A.

Perhaps unsurprisingly, force-feeding the microbial suspensions S1 and S2 did not impact the same genes, except for the insertion sequence *is6100*. For example, the S1 suspension successfully reduced the abundance of *aphA3* and *ermB* at weaning, whereas the S2 suspension did not, even though it harbored similar levels of these genes (Appendix A). Both suspensions also increased two to five different ARGs at weaning (*ermF*, *lnuB*, *mpmB*, *tetQ*, *tnpA* and *vanTG* vs. *lnuC* and *tetW*). While undesired, this increase in ARGs in the force-fed piglets supports the hypothesis that inoculating an anaerobic microbial suspension resulted in foreign microbial species colonizing the gut of the piglets. In total, the abundance of eight and two genes in the feces of piglets at weaning were modified by the inoculation of the microbial suspensions S1 and S2, respectively (*aphA3, ermB*, *ermF*, *lnuB*, *mpmB*, *tetQ*, *tnpA* and *vanTG* in Table 1; *lnuC* and *tetW* in Table 2).

### 3.3. Oral Inoculation of Microbial Suspension Temporarily Affected the Antibiotic Resistance of Enterobacteriacae

Force-feeding the microbial suspension temporarily changed the abundance of the *Enterobacteriacae*, which were isolated from the feces of piglets and subsequently tested for their sulfonamide and tetracycline resistance. Surprisingly, the proportion of resistant *Enterobacteriacae* was higher in the force-fed piglets at 2 weeks of age, even though the donor sows had less resistant *Enterobacteriacae* than the lactating sows at the beginning of the experiment. Unfortunately, the proportion of tetracycline-resistant *Enterobacteriacae* in the donor sows increased from 6 ± 8% to 50 ± 28% during the experiment (Appendix A), and ended up being comparable to the lactating sows at the end of the experiment (48 ± 37%) even though no antibiotics were used. The impact of force-feeding on resistant *Enterobacteriacae* disappeared by the time of weaning (Appendix A), at which point only the impact of the lactating sow remained significant (Appendix A). 

### 3.4. Oral Inoculation of Microbial Suspension Affected the Vertical Transmission of Bacteria

Inoculating the S1 or S2 suspensions affected the fecal microbiota of the piglets collected at weaning (4 weeks). Indeed, the discriminant analysis on the principal components that combined the samples from Days 14 and 27 revealed a significant impact of both inocula compared to the control piglets (*p* = 0.036 and 0.047, based on 1000 random permutations; Figure 1).

To determine if the inoculation resulted in OTUs that are more frequent in piglets than expected by random colonization, the frequency of each OTU was compared to its mean abundance in piglets exposed to the same microbial metacommunity, i.e., applying the neutral model to the piglets sharing the same treatment (S1, S2 or saline) and the same litter. Interestingly, the neutral model describes the species distribution relatively well for control piglets under each sow (R^2^ = 0.28 ± 0.09), but never for S1 or S2 piglets (R^2^ = 0.08 ± 0.15 and 0.04 ± 0.13) (Figure 2 and Appendix A), confirming that a different dynamic is taking place when piglets are orally inoculated. The poor fit of the neutral model implies that the community assembly rules are different when feeding the microbial suspensions S1 or S2 instead of feeding saline water. The goodness-to-fit of the model was not dependent on the age of the piglets (*p* = 0.21, according to ANOVA). It should also be noted that the model was adjusted from an extrapolated ‘piglet metacommunity’ because the microbes naturally colonizing the gut of the piglets are different from the communities in the sows (*p* = 0.001, according to ADONIS), hindering the direct use of the neutral model. In fact, 73% of the OTUs that were always present in the control piglets were undetectable in their lactating sows (Appendix A). Perhaps unsurprisingly, the OTUs relevant to the discrimination of the inoculated piglets are often undetectable or rare in the donor sows (Appendix A). For example, OTU12 is the OTU that contributes the most to the separation between the control, S1 and S2 piglets on the second axis (Figure 1), and is overrepresented in S1 piglets (*p* = 0.004, according to a Wilcoxon test) but merely represented less than 0.02% in the donor sows (Appendix A). 

### 3.5. Relationships between OTUs and ARGs

In order to pinpoint the OTUs carrying or excluding ARGs, the maximal information coefficient index was used to evaluate the linear and co-exclusion relationships between OTUs and ARGs. To avoid capturing relationships due to the impact of age, data at both dates were analyzed separately. At 14 days, the only three significant relationships involving ARGs were OTU23 (*Bacteroides*)-*catA1* (non-coexistence), OTU49 (*Roseburia*)-*cmlA1* and OTU6 (*Bacteroides*)-*cepA*. Many more relationships were detected at weaning (Table 3), at which point the maximal information coefficient was significant for 852 pairs among the 31,626 pairs analyzed, which included 532 linear correlations, 149 co-exclusion relationships and 171 other significant relationships (Appendix A). We observed the OTU6 (*Bacteroides*)-*cepA* relationship again, along with other single linear OTU–ARG correlations pointing towards probable ARG carriers: *lnuB*, *tetG*, *tetM* and *ermG* were linked to OTU43 (*Turicibacter*), OTU198 (*unclassified Bacteroidia*), OTU35 (*Lactobacillus*) and OTU5 (*unclassified Ruminococcaceae)*, respectively. The *tet32* gene was linearly linked both to OTU7 and OTU4458, which share 99% identity and 100% query coverage with the uncultured clone p-4570-4Wb3 found in pigs [35]. Interestingly, we also identified OTUs that never co-exist with specific ARGs, suggesting a possible exclusion mechanism of the microbe carrying the ARGs by the OTU. For example, we observed OTUs connected to numerous ARGs by co-exclusion, such as OTU229 (*unclassified Bacteroidales*) and OTU868 (*Clostridium_sensu_stricto*), which never co-exist with the *aadE*-*aphA3*-*ermG*-*ermB* cluster (Appendix A). The antibiotic resistance genes tend to cluster together, with up to eight different ARGs in the cluster containing *ermG* and *aphA3* (Appendix A), but the clusters are not always associated with OTUs. For example, *mefA* and *tetQ* form a cluster of ARGs with no linear relationship to any of the OTUs, in contrast with the cluster of *tet40*, *ermG*, *InuC*, *aadE*, *tetW*, *aphA3* and *ermB* that includes several OTUs. As expected, the measurements using different primers for the same gene (such as *aphA3* and *aphA3*) cluster together. The OTUs also form clusters containing two to 35 OTUs (data not shown).

## 4. Discussion

This study revisits the concept of competitive exclusion first explored by Nurmi [15]. We demonstrate that the abundance of specific bacterial species is modified by the repeated inoculation of a microbial suspension, which in turn impacts the abundance of antibiotic resistance genes. The success of the ARG exclusion undoubtedly depends on the composition of the inoculated community, since inoculating the S1 and S2 suspensions had markedly different results for ARG and 16S rRNA genes in piglets.

### 4.1. Promoting Sow-to-Piglet Transmission

In our experiment, the piglets were in continuous microbial exposure with their lactating mother through the environment and the nursing. Despite this contact, force-feeding 2 mL of an anaerobic microbial suspension three times a week from Day 2 to Day 23 was enough to modify the gut microbial communities so that the species distribution would no longer fit the neutral model. Thus, the microbial imprint acquired through birth or early investigation of the environment can be overridden by a relatively modest but early inoculation of microbes, even when piglets are not raised in isolators. Several studies excluded the maternal contact from their offspring. For example, piglets in isolators were “humanized” by injecting 3 mL of human fecal suspension [36]. Moreover, piglets in a non-sterile room fed with an automatic feeding system and inoculated with complex microbial communities clustered together [37], suggesting that the inherited microbiota may be replaced by force-feeding the young animals with a fecal suspension. However, these studies ignored the maternal contact even though radioactivity fed to lactating sows can be retrieved in their feces and their piglets, leading to an estimated ingestion of 8.5 g to 20.9 of feces and bedding per day [38]. The contribution of this ingestion of feces to the colonization of the gut is corroborated by the fact that the succession and stabilization events reported in naturally-reared outdoor pigs does not occur when the regular immigration of microbes is shut down by rearing the piglets in isolators, with no contact with the feces of their mother [39]. More recently, inoculation of piglets that still live under their mother showed that changing the microbiota did not require the elimination of all other microbial contacts to be effective: feeding the cross-bred piglets with a fecal suspension derived from another breed resulted in a microbial community enriched in five microbial species [40]. The importance of early-life microbial contact is also illustrated by the ease with which microbes can be inoculated in newly-hatched chicks [17,41]. Taken together, these results show that a complex microbe community free of pathogenic bacteria could be implanted by appropriate exposure of the young animals to complex anaerobic microbial communities. A similar observation was made in rabbits [42]. We confirm here that this tool can also be used in piglets living under their mothers to change the rules that underlie microbial assembly in piglets, as shown by the failure of the neutral model to capture the species distribution in force-fed piglets.

The age at which the piglets are susceptible to their surrounding microbial communities is still unknown, but the pre-weaning period seems to be the most well-adapted: (a) co-housed piglets suddenly develop very similar communities at around 2–3 weeks of age [14]; (b) feeding yeast to piglets before weaning impacts the average daily weight gain of the piglets, whereas feeding yeast after weaning does not [43]; (c) feeding a complex community to piglets between 10 and 18 days of age changed the microbial community in a way that increased robustness when the piglets are submitted to an early weaning at 21 days [40]; and (d) cross-fostering an obese typical Chinese piglet breed and a lean Western breed demonstrated the impact of the nursing mother on the piglets’ microbiota and interleukin 10 [44]. In other species, the microbial environment in early life is also more important than the microbial imprint at birth, as revealed by the cross-adoption of 1-day-old rabbits [13]. Humans may appear as the exception to the rule since babies delivered by C-section still had less vaginal bacteria after 1 month [45], but the vaginal signature represents only a small fraction of the microbial community. For the record, pigs are still mildly susceptible to their microbial environment after weaning, as illustrated by the impact of poor hygiene on microbiota of 8-week-old pigs [27]. Late susceptibility is also observed in humans where competitive exclusion against *Clostridium difficile* was successful in human adults when ingesting encapsulated microbiota [20]. At any rate, exclusion against maternal ARGs in piglets has to be performed after birth and before weaning to limit the diffusion of ARGs on the farm. Taken together, these results suggest that force-feeding piglets could counter the drift of the microbiota exposed to antibiotics after birth [46]. The transmission of antibiotic-resistant microbes to piglets that we observed might also partly explain why conventional and organic farms had similar ARG levels [2]. Indeed, some ARG-carrying microbes (such as *Megasphaera elsdenii*) might arise in piglets after weaning, even if their pen is in a separate room from their mother and no antibiotics are used [47], so that organic farms may carry over ARGs when they introduce piglets raised under ARG-carrying sows.

### 4.2. The Impact of Anaerobic Microbes on Antibiotic Resistance Genes

Initially, none of the ARGs were significantly depressed in the two donor sows compared to the four lactating sows, possibly because of the heterogeneity observed in the four lactating sows, which also translated in piglets with contrasted ARG abundance backgrounds. For example, the control piglets of Sow 512 had up to 80-fold less *mphB* than the control piglets from Sow 722 (Appendix A). Nevertheless, the piglets fed with the selected communities had more resistant *Enterobacteriacae* at Day 14 than the controls, as well as different communities at weaning, showing that the inoculation had an impact, albeit not always a desirable one. Interestingly, the exclusion of the probable ARG carrier could be caused by microbes that were phylogenetically distant. For example, our results showed that OTU5 (*unclassified Ruminococcaceae*) is probably the carrier of *ermB*, which is favored by tylosin when used as a growth promoter in pig production [48].

The largest ARG cluster is *the tetO-tet40-aadE-aphA3-tetW-ermG-ermB-lnuC* cluster. The *ermB* gene was previously described with *aadE* and *aphA3* in *Enterococcus faecium* from poultry manure and sewage [49]. Interestingly, *ermB* is “non-coexistent” with microbes such as OTU229 (*unclassified Bacteroidales*) and OTU868 (*Clostridium_sensu_stricto*). OTU229 shares 99% identity with the clone T1WK15A found in Canadian piglets whose *ermB* abundance did not increase upon exposure to antibiotics [50], in contrast to previous observations [48]. A larger study involving 681 Danish pigs at slaughter showed a correlation between *ermB* and their lifetime exposure to tetracycline, but the model was only able to explain 42% of the variance [51], suggesting that other factors play a role in the *ermB* response upon antibiotic exposure. It can only be speculated as to whether or not one of these factors is the composition of the maternal microbiota, which could explain the high background resistance that is sometimes observed in non-medicated pigs [52].

Exclusion led by microbes from a taxonomic order different from the ARG carrier has practical implications. Indeed, OTUs excluding *ermB* do not belong to the *Ruminococcaceae* family that includes OTU5, the probable carrier of *ermB*. Similarly, an experiment in mice showed that *Barnsiella* (*bacteroidales*) prevents the colonization from vancomycin-resistant *enterococcus* (*lactobacillales*) [18]. The practical implication is that replacing an ARG-resistant microbe by its sensitive kin may not be the optimal strategy. In fact, the inoculation of sensitive *Megasphaera elsdenii* to exclude the resistant strain was not visible at Day 45 despite six inoculations between Day 3 and Day 34 [47].

It should be noted that maintaining anaerobic conditions probably increases the chances of a successful inoculation. For example, anaerobic microbes from the complex community are more efficient at excluding the vancomycin-resistant *enterococcus* than their aerobic kin in mice. Previous successful experiments of fecal transplants in piglets also maintained anaerobic conditions [40]. In our opinion, maintaining anaerobic conditions like those we described in the Materials and Methods section may be the key to success since anaerobes become dominant in piglets after 24 h, reaching 3 × 10^9^ CFU/g [53].

Surprisingly, the large ARG clusters were similar to the clusters observed in China, even though Chinese and French pigs have country-specific ARG profiles [1]. For example, the *dfrA1-sul2-intl1-aadA-aadA2-strB* cluster was also reported in China, as was the association between *mefA* and *tetQ* and between *aphA3* and *tetW* [54]. Conversely, our results showed striking differences with Chinese data that cannot be explained by technical aspects. Indeed, *sul2* is not directly connected to any OTU in our dataset. Even when lowering our threshold to 1%, *sul2* becomes connected to OTU51 (*unclassified Clostridiales*), whereas it is linked to a member of the *Xanthomonadales* in China. Such differences in the likely bacterial host for *sul2* might arise from the mobility of this genetic element, which has been found in very diverse bacteria, ranging from the *Pseudomonadales* order (*Acinetobacter*) to the *Enterobacteriales* order (*Proteus mirabilis*) [54]. Mobile genetic elements may in fact underpin several associations observed in the clusters. For example, the *ermB*-*aadE* and *ermB* -*aphA3* associations are observed on Transposon5385 and Transposon1545 [55] that may be transferred between bacteria, and were reported in 33 different genera [56]. In this context, inoculating complex communities rather than single strains could be preferable because it may simultaneously exclude all the species carrying the transposon and, consequently, hinder the transmission of mobile genetic elements carrying genes such as *ermB*.

The smaller ARG–OTU clusters validate the method used in this study. For example, the OTU6 identified as a probable carrier for *cepA* shares 100% identity with the *Bacteroides fragilis* strain NCTC 9343, which carries a chromosomal beta-lactamase *cepA* [57]. Similarly, OTU35 associated with *tetM* in our study is identical to *Lactobacillus salivarius* JCM1046 whose genome harbors a single copy of an integrated conjugative transposon with the tetracycline resistance gene *tetM* [58].

## 5. Conclusions

In conclusion, feeding complex microbial communities may be advantageous for excluding clusters of antibiotic-resistant genes in young piglets, but great care should be given to avoid an accidental increase in ARGs, as highlighted in this study. This is challenging because the final concentration of ARGs could not be predicted from the simple absence of the target in the donor sow. Further work is therefore needed to improve our understanding and our predictions of competitive exclusion.

## Figures and Tables

**Figure 1 microorganisms-08-01576-f001:**
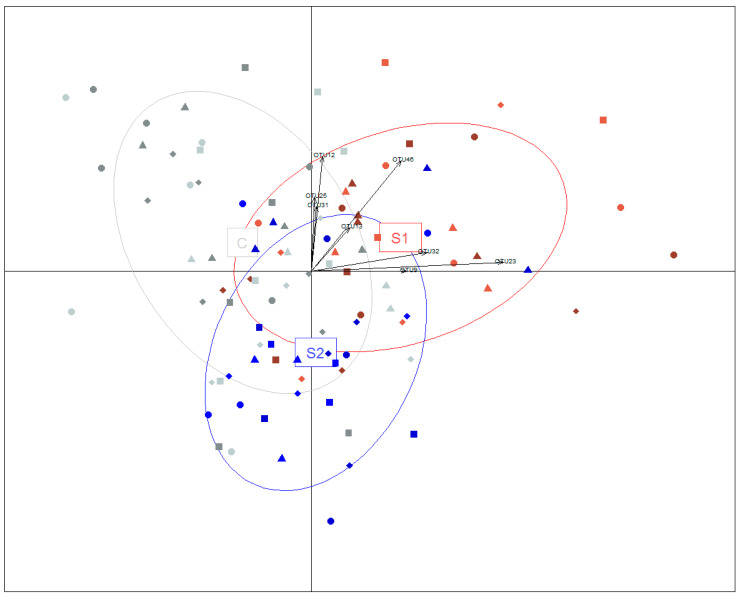
Multidimensional representation of the microbiota of piglets inoculated with either suspension S1, suspension S2 or the control piglets inoculated with saline water. The inset represents the variability taken into account by the principal component analysis (91%). The control piglets are in gray while the piglets force-fed with the S1 or S2 microbial suspensions prepared from the donor sows with low ARG levels are in red and blue. The contributions of the eight operational taxonomic units (OTUs) of interest are indicated by arrows. Piglets from sows 512, 709, 710 and 722 are represented as squares, circles, triangles and diamonds, respectively. The samples at 14 days are shown in lighter shades and the samples at weaning are shown in darker shades.

**Figure 2 microorganisms-08-01576-f002:**
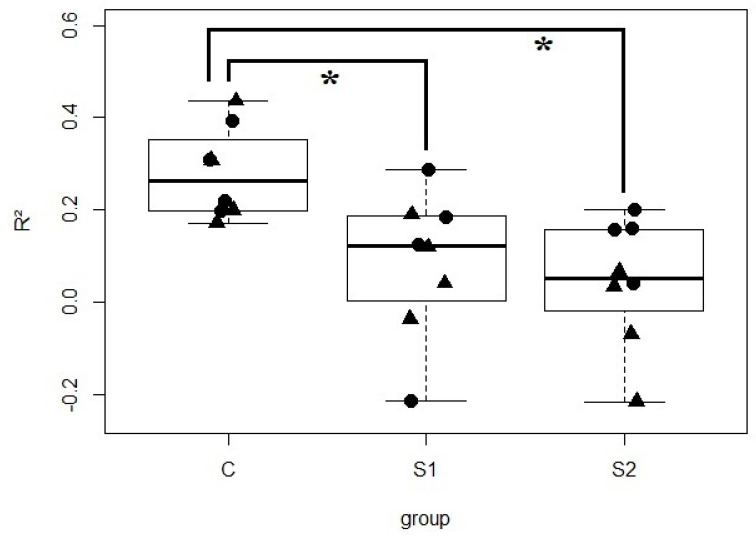
R^2^ values of the neutral model fitting the S1, S2 and control piglets. Circles are the 14-day-old piglets and triangles are the piglets at 4 weeks (weaning). Star (*) indicate that the R^2^ values are significantly different (at the 5% threshold of the ANOVA) between the control piglets and the piglets force-fed with the S1 or S2 microbial suspensions prepared from the donor sows with low ARG levels, illustrating that the inoculation modified the assembly rules of the microbes.

**Table 1 microorganisms-08-01576-t001:** Impact of the suspension from donor sow 268 (S1) on the abundance of antibiotic resistance genes (ARGs) and related genes in the piglets at 14 days and at 4 weeks (median values). The ARG abundances are normalized by the abundance of the 16S rRNA gene. The numbers are not between 0 and 1 because the ARGs were pre-amplified, whereas the 16S rRNA genes were not (see Materials and Methods section, Appendix A and Appendix A). The ARGs *aphA3* and *ermB* are underlined because they are decreased by the inoculation of the microbial suspension S1.

ARG	At 14 Days								At 4 Weeks (Weaning)								P_fried_14d	P_fried_4w
	Sow 512		Sow 709		Sow 710		Sow 722		Sow 512		Sow 709		Sow 710		Sow 722			
	S1 Piglets	Control Piglets	S1 Piglets	Control Piglets	S1 Piglets	Control Piglets	S1 Piglets	Control Piglets	S1 Piglets	Control Piglets	S1 Piglets	Control Piglets	S1 Piglets	Control Piglets	S1 Piglets	Control Piglets		
*aphA3*	24	53	45	14	24	13	51	17	1.4	1.9	2.1	2.5	1.6	3.0	2.9	3.4		*
*ermB*	0.8	1.9	0.3	0.3	0.3	0.4	1.1	0.7	1.6	1.8	3.0	3.4	2.3	3.1	2.5	3.5		*
*aadE_2*	4.1	11.9	4.0	7.6	1.0	2.4	6.3	6.9	2.1	2.2	2.9	2.9	2.2	3.0	1.8	2.7	*	
*cmlA1*	439	19	62	0	801	501	0	0	NA	NA	NA	NA	NA	NA	NA	NA	*	
*dfrA16*	0.3	0.2	0.1	0.0	0.3	0.1	0.2	0.0	NA	NA	NA	NA	NA	NA	NA	NA	*	
*ermF*	0.3	0.1	0.4	3.6	0.5	0.2	0.5	0.2	0.4	0.3	5.1	0.8	1.3	1.1	1.1	0.2		*
*IntI1*	61.7	18.2	9.0	0.1	111.0	54.4	23.3	9.0	4.0	8.9	63.7	7.0	3.5	2.8	28.5	12.5	*	
*is6100 ^a^*	2	21	75	417	1	2	55	574	1.2	2.3	1.1	0.8	1.6	2.6	1.0	1.4	*	
*lnuB*	0.29	0.00	0.01	0.00	0.04	0.04	0.07	0.03	3.4	1.5	1.9	1.0	1.0	0.8	1.3	1.0	*	*
*mphB*	38.2	0.2	0.1	1.0	0.0	0.0	0.3	0.2	6.5	0.4	359.5	21.9	1.5	0.3	206.2	33.6		*
*mphE*	4.1	2.5	11.4	1.4	0.4	0.3	2.4	0.6	NA	NA	NA	NA	NA	NA	NA	NA	*	
*npmA*	4088	1283	4819	6	492	15	4351	2298	5.7	6.2	4.4	4.9	19.3	9.0	45.3	57.1	*	
*tetQ*	0.2	0.1	0.2	0.7	0.1	0.1	0.1	0.0	0.6	0.5	1.5	1.0	1.0	0.9	0.5	0.3		*
*tnpA ^b^*	0.008	0.01	0.02	0.003	0.001	0.001	0.003	0.03	2.1	0.8	0.3	0.1	1.4	0.9	0.5	0.3		*
*vanTG*	0.27	0.05	0.01	0.004	0.01	0.01	0.07	0.005	0.22	0.05	0.70	0.2	0.21	0.07	0.46	0.03	*	*

^a^: Insertion sequence, not an ARG. ^b^: Transposase, not an ARG, *statistically significant à the 5% threshold.

**Table 2 microorganisms-08-01576-t002:** Impact of the suspension from donor sow 206 (S2) on the abundance of ARGs and related genes in the piglets at 14 days and at 4 weeks (median values). The ARG abundances are normalized by the abundance of the 16S rRNA gene. The numbers are not between 0 and 1 because the ARGs were pre-amplified, whereas the 16S rRNA genes were not (see Materials and Methods section, Appendix A and Appendix A). The ARG *tetL* is underlined because it is decreased by the inoculation of the microbial suspension S2 of the piglets.

Primer Set ID	At 14 Days								At 4 Weeks (Weaning)								P_friedman_14days	P_friedman_4weeks
	Sow 512		Sow 709		Sow 710		Sow 722		Sow 512		Sow 709		Sow 710		Sow 722			
	S2 Piglets	Control Piglets	S2 Piglets	Control Piglets	S2 Piglets	Control Piglets	S2 Piglets	Control Piglets	S2 Piglets	Control Piglets	S2 Piglets	Control Piglets	S2 Piglets	Control Piglets	S2 Piglets	Control Piglets		
*tetL*	0.001	0.011	0.0001	0.0004	0.003	0.007	0.003	0.018	0.05	0.05	0.02	0.78	1.90	1.77	0.01	0.03	*	
*aac6Im*	66.95	0.24	0.02	0.01	0.76	0.15	2.33	0.24	NA	NA	NA	NA	NA	NA	NA	NA	*	
*aadA*	311.85	69.3	54.75	51.18	465.4	83.50	12.08	4.26	0.28	0.19	0.15	0.16	0.40	0.08	0.62	0.22	*	
*aph2Ib*	5217.6	7.2	1.3	1.1	69.9	10.8	201.7	17.7	167	179	218	33.2	14.9	3.5	147	129	*	
*dfrA16*	0.56	0.19	0.07	0.04	0.14	0.12	0.23	0.03	NA	NA	NA	NA	NA	NA	NA	NA	*	
*is6100 ^a^*	65.36	21.3	1257.7	416.8	2.78	2.37	643.2	574.4	3.56	2.31	2.15	0.82	1.43	2.58	1.25	1.36	*	
*lnuC*	2.83	1.74	0.50	1.68	1.06	1.00	1.80	1.12	1.87	1.03	1.62	1.07	1.47	0.86	1.12	1.00		*
*tetW.1*	0.45	0.27	0.24	0.21	0.17	0.11	0.25	0.20	2.08	1.03	2.39	1.47	1.25	0.93	1.78	1.28	*	*

^a^: Insertion sequence(not an ARG), * statistically significant à the 5% threshold.

**Table 3 microorganisms-08-01576-t003:** Remarkable links between OTUs and ARGs at weaning.

Type of Relationship	ARGs or IS	Targeted Antibiotics	Linked OTU	Taxonomy of the Linked OTU
Linear relationship: proportionality between OTU and ARG abundances	*aphA3*	Kanamycin	131	*unclassified Clostridia*
			2302	*unclassified Clostridiales*
	*cepA*	Cephalosporin	6	*Bacteroides fragilis*
	*tetG*	Tetracycline	198	*unclassified Bacteroidia*
	*npmA*	Aminoglycoside	10	*unclassied Clostridiales*
			12,054	*unclassied Clostridia*
	*tetM*	Tetracycline	35	*Lactobacillus*
	*lnuB*	Lincosamide	43	*Turicibacter*
	*ant6-Ib*	Streptomycin	28	*unclassified Fusobacteriaceae*
			120	*unclassified Clostridiales*
	*dfrA1-sul2-intl1-aadA-aadA2-strB*	Diaminopyrimidine-Sulfonamide-Aminoglycoside		
	*blaTEM- tetR- tetA*	β-lactam antibiotics-Tetracylcline		
	*mefA-tetQ*	Macrolide-Tetracycline		
	*tetO-tet40-aadE-aphA3-tetW-ermG-ermB-lnuC*	Tetracycline-Streptomycin-Kanamycin-Macrolide-Lincosamide-Streptogramin B		
	*tet32*	tetracycline	7	*unclassified Clostridia*
			4458	*unclassified Firmicutes*
	*Intl1*	-	51	*unclassified Clostridiales*
	*ermB*	Macrolide-Lincosamide-Streptogramin B	5	*unclassified Ruminococcaceae*
Co-exclusion relationship: the gene was never observed together with the OTU	*ermB and ermG*	Macrolide-Lincosamide-Streptogramin B	229	*Unclassified Bacteroidales*
			100	*Unclassified Bacteroidales*
			193	*Prevotella*
			868	*Clostridium_sensu_stricto*
	*tetL*	Tetracycline	57	*unclassified Firmicutes*
			12	*unclassified bacterium*
	*ermF*	Macrolide-Lincosamide-Streptogramin B	5858	*unclassified Clostridia*

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
