# Peer review of "Early Inoculation of Microbial Suspension in Suckling Piglets Affects the Transmission of Maternal Microbiota and the Associated Antibiotic Resistance Genes"

_microorganisms, 2020, doi:10.3390/microorganisms8101576_

Round 1
Reviewer 1 Report
Minor issues:
- In the abstract the genes named for the first time should be spelled out.
- There is no relevance given that justifies the experiment. Why is artificial transfer of the microbiome of adults sows an appropriate tool?
- What is the difference between S1 and S2?
- In the introduction the authors should explain in more detail which ARGs are country specific and which antibiotics that are used in humans are affected.
- Fecal transplants are the one and only cure for C. difficile.
- Were the donors selected only on the base of the abundance of Enterobacteriaceae?
- The authors should show the distribution of Enterobacteriaceae in the whole herd.
- Why did the authors test only for ARGs of tetracycline and sulfamethoxazole?
- The display of table 1 should be in a graph.
Major issues:
- The concept that fecal transplants from adult pigs leads to outcompeting ARG carrying bacteria in piglets is not convincing, because adults, which have seen antibiotics already some when in their lifetime should carry more ARGs than piglets.
- There was no difference in ARGs between the donors and the lactating sows. This is in contrast to the initial concept and hampers all conclusion on reduction of ARGs by admitting S1 or S2.
The concept of outcompeting ARG carrying bacteria by fecal transplant is an interesting one but requires better control of the starting conditions.
Author Response
Answer to the 1st review of the paper no. microoranisms-916340
Minor issues:
- In the abstract the genes named for the first time should be spelled out.
This was done.
- There is no relevance given that justifies the experiment. Why is artificial transfer of the microbiome of adults sows an appropriate tool?
Piglets are usually colonized by the fecal microbes of their mother by ingesting its feces : In fact, radioactivity fed to lactating sows can be retrieved in their feces and their piglets, leading to an estimated ingestion of 8.5 g to 20.9 of feces and bedding per day(L350). Hence, we aimed to mimic this behavior.
From an applied point of view, it is also interesting to select an adult donor because a sow could be kept completely off antibiotic to be used as a donor in each farm, which alleviates regulatory issues and risk management that arise when importing complex microbial communities in a farm (quarantine, etc…)
- What is the difference between S1 and S2?
S1 and S2 originate from two different sows because we had prior experience with inoculating complex microbial communities to rabbits, where we observed a large variability in the ability to exclude ARGs. Therefore we did not want to put all our eggs in the same “donor basket”.
Variability of the Ability of Complex Microbial Communities to Exclude Microbes Carrying Antibiotic Resistance Genes in Rabbits.
Achard CS, Dupouy V, Siviglia S, Arpaillange N, Cauquil L, Bousquet-Mélou A, Zemb O.
Front Microbiol. 2019 Jul 2;10:1503. doi: 10.3389/fmicb.2019.01503. eCollection 2019.
- In the introduction the authors should explain in more detail which ARGs are country specific and which antibiotics that are used in humans are affected.
This was done.
- Fecal transplants are the one and only cure for C. difficile.
The text was changed into:” Nowadays, human fecal transplantation using complex microbial communities can cure Clostridium difficile infections [19]”
- Were the donors selected only on the base of the abundance of Enterobacteriaceae?
In addition to test the abundance of Enterobacteriacea, we did check that some variability of ARGs was present in the herd, as now shown in the supplemental data by the following graph :
- The authors should show the distribution of Enterobacteriaceae in the whole herd.
We did not measure the distribution of Enterobacteriaceae in the whole herd, and we are unsure how this rather time consuming measure would have helped.
- Why did the authors test only for ARGs of tetracycline and sulfamethoxazole?
We measured 28 ARGs by qPCR. We focused on tetracycline- and sulfamethoxazole- resistant enterobacteria by cultural methods because these antibiotics are found in liquid pig manure used as fertilizer (Hölzel et al, 2010), so it was interesting for us to see if we could impact the tetracycline- and sulfamethoxazole- resistant enterobacteria.
- The display of table 1 should be in a graph.
We feel that a graph is harder to read in this instance, so we add the requested graph in the supplementary data (Figure S5 and S6)
Major issues:
- The concept that fecal transplants from adult pigs leads to outcompeting ARG carrying bacteria in piglets is not convincing, because adults, which have seen antibiotics already some when in their lifetime should carry more ARGs than piglets.
It is true that using bacteria carried by piglets would have made sense, since they have probably have been less exposed to antibiotics. However our rationale to use bacteria carried by adults was that we wanted to mimic the natural situation, in which piglets colonize themselves by ingesting 8.5 g to 20.9 of maternal feces as shown by radioactivity fed to lactating sows (L350). Therefore we picked sows with low levels of ARGs as donors.
From an applied point of view, it is also interesting to select an adult donor because a sow could be kept completely off antibiotic to be used as a donor in each farm, which alleviates regulatory issues and risk management that arise when importing complex microbial communities in a farm (quarantine, etc…)
- There was no difference in ARGs between the donors and the lactating sows. This is in contrast to the initial concept and hampers all conclusion on reduction of ARGs by admitting S1 or S2.
First, we wish to emphasize that there is a difference in the proportion of resistant enterobacteria between the donors and the lactating sows. The difference in ARG is not significant for 2 reasons in our opinion:
- We measured many ARGs, so that the correction for the false discovery rate makes it harder to obtain a difference
- We concentrate the sampling effort on the piglets, where the impact is expected, so that the statistical power on the donors vs the lactating sows is relatively low
- As illustrated in the Figure S2, there is no obvious relationship between the ARG measured in the donors, the ARG measured in the sows, and the ARGs measured in the piglets. In our opinion, the simple hypothesis that inoculating with a low level of ARG X should result in piglets with decreased ARG X is not supported by the data, as discussed in the MS (L240) : “The difficulty of predicting the success of the competitive exclusion is striking for the 16S rRNA methyltransferase npmA against aminoglycosides, which was present in each lactating sow, undetectable in the S1 and S2 suspensions and, yet not modified by force-feeding the suspensions (p=0.32 according to the Wilcoxon test) (Tables S2 and S3). The complexity of predicting the success of the ARG exclusion from the ARGs in the donor and the lactating sow is presented in Fig. ”
The concept of outcompeting ARG carrying bacteria by fecal transplant is an interesting one but requires better control of the starting conditions.
This comment refers to the absence of difference between donors and lactating sows according to the reviewer as mentioned above, yet donors and lactating sows did differ significantly in their proportion of resistant enterobacteria. Therefore this comment is not entirely relevant in our opinion for the reasons explained above.

Reviewer 2 Report
Review of the paper no. microoranisms-916340
The results of this experiment have cognitive elements and great importance for pig producers and pork consumers. The aim of this study was to evaluation the effect of oral early inoculation of fecal microbial suspension on antibiotic resistance of microbes (ARG) in piglets. The number of sows and piglets used in the experiment is sufficient, the test methods used are correct. The discussion is well carried out and exhausting. References well chosen. The numbering of References in the Materials and Methods, Results and Discussion chapters requires revision. For the References chapter it is required to use the appropriate font (see template on the journal website) and abbreviated name journals. Before publishing in Microorganisms, the paper requires additions and corrections. The list of proposed changes is given below:
L8-11 provide names of research units in English, full names for INRAE, INPT, ENVT, INTHYERES, PEGASE
L8-11 provide email addresses for all authors of the article, initials used for the Author Contributions part,
L30 + please enter a space before the References numbers in the text of the article
L50, please provide information on the acquisition of resistance to therapeutic antibiotics by some bacteria in pigs, pig production programs without antibiotics (including therapeutic) e.g. by Goodvalley, the impact of discontinuation of feed antibiotics on the health (intestinal health) and productivity of pigs.
L66 INRAEA UE3P what does that mean?, give the full name
L66 + please provide information about the type of building (closed, with or without windows?), size of the pen, type of floor, microclimate conditions (temperature from to), humidity, concentration of harmful gases, length, type, light intensity during the test period for sows and piglets, information about prophylactic procedures (iron injection, sprout cutting, curling the tails, etc.)
L94 [46] should be [20], in the Materials and Methods chapter, the consecutive numbers from 20 should be used, not from 46 to 54, the numbering in lines 94, 121, 131, 134, 142, 146, 152, 162, 171, 177 needs to be corrected
L143 SPR-12492 must be in Palatino Linotype, font 10
L207 “van TG and LnuB” were increased by the S1 suspension at 14 days also?
L230 Fig S1, space after Fig.
L233 aph3 and Ern B are present in S2 suspension?
L210, 219 or 14 days, In Line 91 is 16 days?
L318 [10] after Nurmi, instead of “in broilers”
L318 + space between References and the preview word
L433 There must be a conclusion regarding my own (this) research, without quoting other authors, without "[26}"
L455 + must be Palatino Linotype 9, abbreviated name of References no 1, 5, 8, 11, 13, 14, 22, 30, 33, 37-40, 42, 44, 53.
Author Response
Answer to the 2nd Review of the paper no. microoranisms-916340
The results of this experiment have cognitive elements and great importance for pig producers and pork consumers. The aim of this study was to evaluation the effect of oral early inoculation of fecal microbial suspension on antibiotic resistance of microbes (ARG) in piglets. The number of sows and piglets used in the experiment is sufficient, the test methods used are correct. The discussion is well carried out and exhausting. References well chosen. The numbering of References in the Materials and Methods, Results and Discussion chapters requires revision. For the References chapter it is required to use the appropriate font (see template on the journal website) and abbreviated name journals. Before publishing in Microorganisms, the paper requires additions and corrections. The list of proposed changes is given below:
L8-11 provide names of research units in English, full names for INRAE, INPT, ENVT, INTHYERES, PEGASE
The full names were provided below but the university asks us to keep the abbreviated addresses for their reference softwares.
GenPhySE : Génétique Physiologie et Systèmes d'Elevage
INRAE: Institut national de recherche pour l'agriculture, l'alimentation et l'environnement
INP Toulouse: Institut National Polytechnique de Toulouse
ENVT: École nationale vétérinaire de Toulouse
INTHERES: Innovations thérapeutiques et résistances
PEGASE: Physiologie, Environnement et Génétique pour l'Animal et les Systèmes d'Élevage
L8-11 provide email addresses for all authors of the article, initials used for the Author Contributions part,
|
full name |
initials |
|
|
C.S. Achard |
CSA |
cachard@lallemand.com |
|
V. Dupouy |
VD |
v.dupouy@envt.fr |
|
L. Cauquil |
LC |
laurent.cauquil@inrae.fr |
|
N. Arpaillange |
NA |
n.arpaillange@envt.fr |
|
A. Bousquet-Melou |
ABM |
a.bousquet-melou@envt.fr |
|
N. Le Floc'h |
NLF |
nathalie.lefloch@inrae.fr |
|
O. Zemb |
OZ |
olivier.zemb@inrae.fr |
L30 + please enter a space before the References numbers in the text of the article
This was done.
L50, please provide information on the acquisition of resistance to therapeutic antibiotics by some bacteria in pigs, pig production programs without antibiotics (including therapeutic) e.g. by Goodvalley, the impact of discontinuation of feed antibiotics on the health (intestinal health) and productivity of pigs.
This was done: “In response to the increase in antibiotic resistance, the European Union banned antibiotics as growth promoters for livestock in 2006, and data suggests that the impact of this ban on the performances of growing pigs is relatively minor at the country-level in Denmark [5] or in carefully controlled groups [6,7]. However, a long-term study conducted at the University of Kentucky showed that stopping antibiotics could also be detrimental to reproductive performance since sows not treated with antibiotics farrowed and weaned fewer pigs per litter [8]. Therefore, excluding antibiotics from the nursery is far from trivial.
As a matter of fact, the 2006 ban on fattening animals might not be sufficient to decrease antibiotic resistance, as seen by Spanish MRSA strains that became more resistant to multiple antibiotics between 2009 and 2018 [9]”
L66 INRAEA UE3P what does that mean?, give the full name
The text was changed: Pig Physiology and Phenotyping Experimental Facility
L66 + please provide information about the type of building (closed, with or without windows?), size of the pen, type of floor, microclimate conditions (temperature from to), humidity, concentration of harmful gases, length, type, light intensity during the test period for sows and piglets, information about prophylactic procedures (iron injection, sprout cutting, curling the tails, etc.)
This information is now available in the updated supplementary material : “The pigs were housed in farming conditions as authorized by the French regulation on animal experimentation for experimentations conducted with an agronomic purpose.
At 106 days of gestation, sows were moved to the farrowing room with natural (windows) and artificial light. They were kept in individual farrowing crates (2 x 2.5 m) equipped with a though and a drinker. The floor of the farrowing pens was made of slatted plastic. The farrowing crates were equipped with 2 infrared heat bulbs for piglets. The ambient temperature was kept at 24 to 25 °C in the lactation rooms with heating and ventilation. Farrowing was induced by an intramuscular injection of prostaglandin F2α (2 mL of Dinolytic, Zoetis, France) on day 114 of gestation. Usual farm practices for newly farrowed piglets were individual identification by tagging, iron injection, tail docking, and castration of male piglets. Cross-fostering, if needed, was performed intra-treatment within 2 days after birth but piglets selected for blood and feces samplings were maintained with her biological mother.”
L94 [46] should be [20], in the Materials and Methods chapter, the consecutive numbers from 20 should be used, not from 46 to 54, the numbering in lines 94, 121, 131, 134, 142, 146, 152, 162, 171, 177 needs to be corrected
This was done. Thank you for the careful proofreading.
L143 SPR-12492 must be in Palatino Linotype, font 10
This was done.
L207 “van TG and LnuB” were increased by the S1 suspension at 14 days also?
This is correct.
L230 Fig S1, space after Fig.
This was done.
L233 aph3 and Erm B are present in S2 suspension?
The levels of these genes were similar in S1 and S2, which is why it is hard to predict the result of the competitive exclusion. The text was changed into : “For example, the S1 suspension successfully reduced the abundance of aphA3 and ermB at weaning, whereas the S2 suspension did not even though it harbored similar levels of these genes (Table S2). »
L210, 219 or 14 days, In Line 91 is 16 days?
This was corrected.
L318 [10] after Nurmi, instead of “in broilers”
The text was changed into : “This work revisits the concept of competitive exclusion first explored by Nurmi”
L318 + space between References and the preview word
A space was introduced before every reference.
L433 There must be a conclusion regarding my own (this) research, without quoting other authors, without "[26}"
This was done. The text was changed into : “In conclusion, feeding complex microbial communities may have benefits in excluding clusters of antibiotic-resistant genes in young piglets, but great care should be given to avoid an accidental increase of ARGs as highlighted in this study.”
L455 + must be Palatino Linotype 9, abbreviated name of References no 1, 5, 8, 11, 13, 14, 22, 30, 33, 37-40, 42, 44, 53.
The police was changed, but I am unsure how I should abbreviate the names.

Round 2
Reviewer 2 Report
Review of the paper no. microoranisms-916340_R1
The results of this experiment have cognitive elements and great importance for pig producers and pork consumers. The aim of this study was to evaluation the effect of oral early inoculation of fecal microbial suspension on antibiotic resistance of microbes (ARG) in piglets. The number of sows and piglets used in the experiment is sufficient, the test methods used are correct. The discussion is well carried out and exhausting. References well chosen. The subsection numbers in the Materials and Methods, Results, and Discussion sections are required. For the References section it is required to use the abbreviated name for all journals (if it is). Before publishing in Microorganisms, the paper requires additions and corrections. The list of proposed changes is given below:
L8-11 provide names of research units in English, full names for INRAE, INPT, ENVT, INTHYERES, PEGASE
L8-11 provide email addresses for all authors of the article, initials used for the Author Contributions part.
L83 + please provide information about the type of building (closed, with or without windows?), size of the pen, type of floor, microclimate conditions (temperature from to), humidity, concentration of harmful gases, length, type, light intensity during the test period for sows and piglets, information about prophylactic procedures (iron injection, sprout cutting, curling the tails, etc.)
Please eliminate the spaces between paragraphs in the text of the article.
The subsection numbers in the Materials and Methods, Results, and Discussion sections are required.
L222 add “(Table 1).” after dates
Tables 1 and 2 without gray in the header.
Table 2 on page no. 6
Abbreviated name of References must be for no. 1, 7, 13, 16. 18, 19, 21, 24, 28, 33, 35, 37, 46, 48, 49, 50, 52, 53, 54, 55, 57, 59.
L486, 597 PLOS ONE or Plos One?
Check compliance with the instructions for authors: no.569-572, 586-588,
Author Response
L8-11 provide names of research units in English, full names for INRAE, INPT, ENVT, INTHYERES, PEGASE
This was done.
L8-11 provide email addresses for all authors of the article, initials used for the Author Contributions part.
This was done.
L83 + please provide information about the type of building (closed, with or without windows?), size of the pen, type of floor, microclimate conditions (temperature from to), humidity, concentration of harmful gases, length, type, light intensity during the test period for sows and piglets, information about prophylactic procedures (iron injection, sprout cutting, curling the tails, etc.)
This is now also mentioned in the main MS (L94-104):”The pigs were housed in farming conditions as authorized by the French regulation on animal experimentation for experimentations conducted with an agronomic purpose. At 106 days of gestation, sows were moved to the farrowing room with natural (windows) and artificial light. They were kept in individual farrowing crates (2 x 2.5 m) equipped with a though and a drinker. The floor of the farrowing pens was made of slatted plastic. The farrowing crates were equipped with 2 infrared heat bulbs for piglets. The ambient temperature was kept at 24 to 25 °C in the lactation rooms with heating and ventilation. Farrowing was induced by an intramuscular injection of prostaglandin F2α (2 mL of Dinolytic, Zoetis, France) on day 114 of gestation. Usual farm practices for newly farrowed piglets were individual identification by tagging, iron injection, tail docking, and castration of male piglets. Cross-fostering, if needed, was performed intra-treatment within 2 days after birth but piglets selected for blood and feces samplings were maintained with her biological mother »
Please eliminate the spaces between paragraphs in the text of the article.
This was done.
The subsection numbers in the Materials and Methods, Results, and Discussion sections are required.
This was done.
L222 add “(Table 1).” after dates
This was done.
Tables 1 and 2 without gray in the header.
This was done.
Table 2 on page no. 6
This was done.
Abbreviated name of References must be for no. 1, 7, 13, 16. 18, 19, 21, 24, 28, 33, 35, 37, 46, 48, 49, 50, 52, 53, 54, 55, 57, 59.
The full names of the journals were replaced by the abbreviated names.
L486, 597 PLOS ONE or Plos One?
This was corrected into : “PLoS One”
Check compliance with the instructions for authors: no.569-572, 586-588,
This was corrected. One of the reference was eliminated because it was redundant.